# Development and Characterization of Hybrid, Temperature Sensing and Heating Yarns with Color Change

**DOI:** 10.3390/s23167076

**Published:** 2023-08-10

**Authors:** Theresa Junge, Rike Brendgen, Carsten Grassmann, Thomas Weide, Anne Schwarz-Pfeiffer

**Affiliations:** 1Research Institute for Textile and Clothing (FTB), Niederrhein University of Applied Sciences, Webschulstr. 31, 41065 Mönchengladbach, Germany; rike.brendgen@hs-niederrhein.de (R.B.); thomas.weide@hs-niederrhein.de (T.W.); 2Faculty of Textile and Clothing Technology, Niederrhein University of Applied Sciences, Webschulstr. 31, 41065 Mönchengladbach, Germany

**Keywords:** flexible sensor, temperature, heating, thermochromic, coating, winding, smart textiles

## Abstract

A person’s body temperature is an important indicator of their health status. A deviation of that temperature by just 2 °C already has or can lead to serious consequences, such as fever or hypothermia. Hence, the development of a temperature-sensing and heatable yarn is an important step toward enabling and improving the monitoring and regulation of a person’s body temperature. This technology offers benefits to several industries, such as health care and sports. This paper focuses on the characterization and development of a hybrid yarn, which can measure and visualize temperature changes through a thermoresistive and thermochromic effect. Moreover, the yarn is able to serve as a flexible heating element by connecting to a power source. The structure of the yarn is designed in three layers. Each layer and component ensures the functionality and flexibility of the yarn and additional compatibility with further processing steps. A flexible stainless steel core was used as the heat-sensitive and heat-conducting material. The layer of polyester wrapped around the stainless steel yarn improves the wearing comfort and serves as substrate material for the thermochromic coating. The resulting hybrid yarn has a reproducible sensory function and changes its resistance by 0.15 Ω between 20 and 60 °C for a length of 30 cm. In addition, the yarn has a uniform and reproducible heating power, so that temperature steps can be achieved at a defined length by selecting certain voltages. The thermochromic color change is clearly visible between 28 and 29 °C. Due to its textile structure, the hybrid sensory and actuating yarn can easily be incorporated into a woven fabric or into a textile by means of joining technology sewing.

## 1. Introduction

Temperature as a physical state variable describes the thermal state of a system [1]. It serves as an important indicator in health care, meteorology, and manufacturing industries. Even minor temperature fluctuations can have a considerable influence on technical processes, the environment, and people [2]. For example, just two degrees deviation of the body temperature from the optimal 36.6 °C can have a negative impact on a person’s health [3,4]. The COVID-19 pandemic, in particular, has once again shown that the measurement of body temperature is usually the first means of medical diagnostic assessment [5]. Less well-known is that low room temperatures of 18 °C, instead of the optimal 24 °C, cause work performance to drop by 5 to 15% [6]. Flexible heating elements can adapt to uneven surfaces and form a higher contact area with the body. The result is the prevention of large heat and energy losses [7]. As flexible heating elements, textiles prove to be an ideal tool for personalized thermoregulation. They are advantageous over traditional central space heating and cooling systems due to energy and space savings, and can serve as a solution for personal thermal management [8,9].

The above-given information indicates the contribution that both a temperature-sensing yarn and a heating function of textiles make to improve everyday life. The optimal solution would be to combine the two functions in a textile product that can switch between the sensory or heating function depending on the situation.

Flexible temperature sensors can be produced through the development of various technologies, such as spinning, weaving, knitting, technical embroidery, as well as printing and coating [10]. Fiber- or yarn-based temperature sensors especially offer high flexibility, good modifiability, processability, and saving of high manufacturing costs, which is why a yarn-based temperature sensor/heating element was produced. They are classified according to the measuring principle as thermoelectric, semiconducting, or thermoresistive [5]. For this paper, the focus was laid on metal resistance thermometers based on a thermoresistive effect due to the simple structure and the flexibility gained from this. Commonly used materials, such as graphene [11], carbon fibers, and carbon nanotubes [12], are used as temperature sensors due to their high electrical conductivity, low cost, and high stability. Materials and polymers, such as conductive particle filled silicone rubber or PEDOT:PSS [13], have also been used to develop highly flexible and stretchable sensors [14]. However, temperature-sensitive metal fibers show the highest precision [15]. Various metallic wires and metal alloy wires, such as copper, nickel, tungsten, stainless steel, and chromium, have been used to develop a flexible temperature sensor by knitting, weaving, and embroidery technologies [16,17,18,19]. To achieve higher mechanical resistance and elongation of the metal fiber, wrap spinning technology has been chosen. Yang et al. developed a flexible temperature sensor consisting of a platinum wire wrapped with polyamide 6.6. The additional layers of multifilament provided higher mechanical resistance and higher elongation of the metal fiber [20].

Furthermore, metal fibers can also be used as a flexible heating element due to their appropriate electrical conductivity [21,22,23]. By connecting a power source, the metallic wire can be heated in a controlled manner. Thus, Hamdani and his team developed two knitted heating elements, consisting of a stainless steel wire and a silver-coated polyester yarn. It was found that the stainless steel heating element can generate a higher amount of heat than the silver-coated polyester yarn [24]. In addition, the fabric of Kayacan and Bulgun could reach temperatures of 128 °C within 10 min by inserting stainless steel wires powered with 12 V [25]. Furthermore, a textile heating element can also be produced by coating technology. For example, fabrics and yarns have already been coated with conductive materials, such as silver nanowires, and heated by a voltage [26,27,28].

The aim of this paper is to combine a temperature-sensing function and a heating function in one yarn, so that the function can be changed according to the application. This dual functionality has received little attention [29,30,31] but opens up a lot of possibilities. Found literature describes the combination of sensing temperature and heating, but those developments are on a fabric level. Here, a hybrid yarn is constructed that comprises sensing and heating and adds a visual feedback mechanism. Sensing and heating are realized by using a metallic conductor. The resistance of this conductor changes due to the increasing movement of electrons when the temperature rises. A linear and reproducible change in resistance is advantageous for sensitivity and evaluation [2]. Due to the conductive properties of the metal, the material can also be used to conduct and transfer heat. With the connection of a voltage source, it is possible to use the conductor as a flexible heating element. The temperature sensing function is supported by a thermochromic color change. The effect of thermochromism describes a change in the color of certain materials when a certain temperature is reached [32]. The thermochromic color change can also be used to identify temperature changes. Thus, the yarn also implements a direct and visual feedback function.

Regarding the lack of moisture and dye adsorption capacity of metals, a substrate material must be wrapped around the conductive core material before coating. The absorption of the thermochromic pigments is taken over by a substrate medium that wraps around the conductive core. The structure can be seen in Figure 1. Accordingly, the yarn is realized by combining different materials and textile technologies to produce a three-layered yarn structure. This yarn structure allows further processing and textile integration with common textile technologies, such as weaving and knitting, but also embroidery and sewing technology.

## 2. Materials and Methods

### 2.1. Materials

The metallic conductor in the core of the yarn and its resistive behavior is decisive for the temperature-sensing and heating function. Depending on the primary use of the sensor, the material must be chosen accordingly. If the heating function is the main objective, an electrical conductor with a moderate specific resistance will be used. Contrarily, in the case of temperature sensing, an electrical conductor that largely changes its resistance upon temperature increase is ideal. This enables the measurement of even small changes in temperature. If both functions—heating and sensing—are to be combined in one, a material must be found that has neither a particularly high nor a low specific resistance.

In order to serve both functionalities, a 100% stainless steel yarn, type Bekinox VN 12/2x275/175S from the Belgian company Bekaert has been selected. It offers a good heating capacity and a resistance of 14.41 Ω/m. For the application of thermochromic pigments, different substrate materials are selected. The substrate has a significant influence on the homogeneity of the coating. Primarily, a 100% cotton rotor spun yarn from Schlafhorst Spinntechnikum (Übach-Palenberg, Germany) is used. Secondly, a rotor spun yarn consisting of 100% polyester from Wagenfelder Spinnereien GmbH (Wagenfeld, Germany) with a fineness of 10dTex is chosen. The third selected yarn is a textured polyester yarn from Saurer (Argon, Switzerland), which has a fineness of 100f333. This selection allows the characterization of how far the fibrous materials, as well as the yarn design, influence the homogeneity of the coating and, consequently, the color change. Used materials and their specific properties are listed in Table 1.

The thermochromic coating consists of a commercially available Tubicoat thickener and polyurethane paste Tubicoat PU60 (CHT Germany GmbH, Tübingen, Germany) filled with thermochromic pigments from Kremer (reference number: 56843). The coating formulation is listed in Table 2 below. According to the manufacturer, the thermochromic pigments change their color from pink to colorless at a temperature of 31 °C.

### 2.2. Methods

Appropriate methods and procedures are described for the production and characterization of the temperature-sensing heating yarn with color change.

#### 2.2.1. Winding Technology

The production of the yarn is realized by a combination of two independently running processes: the winding and the coating process. The winding of the metallic conductor was carried out using a hollow spindle spinning setup, type FantaOne, from the Italian company Gualicheri e Gualicheri. The spinning process is schematically illustrated in Figure 2. The core material is pulled off overhead and guided to the substrate material using pairs of rollers and deflection rollers. The wrapping material is drawn off in a controlled manner by a draw-off roller and rotation of the spindle. Accordingly, the degree of wrapping is determined by the ratio of the take-off speed and the rotation speed. For each material combination, the speeds must be defined to achieve optimal wrapping of the core without overlapping or missing loops. For the production of the hybrid yarn, a constant twist speed of 5000 rpm is used. The degree of twist is adjusted by changing the number of revolutions in tpm (twists per meter). For rough orientation, the speed of 1500 tpm is used and gradually increased until a uniform twist can be seen.

#### 2.2.2. Coating Process of Color Changing Layer

The coating dispersion is applied by a vertical dip coating technology [33,34]. Figure 3 shows the schematic representation of the coating process. The substrate material is unwound vertically from the spool and passes through the nozzle filled with coating dispersion. The openings in the nozzle, at the bottom and top, safeguard a dispersion uptake on the yarn and also homogenize the coating along the yarn. After coating, the yarn is dried and wound up.

Important setting parameters, such as the viscosity of the dispersion, coating speed, and fiber material, must be identified for each material combination and/or yarn design. The following Table 3 describes the tests for the characterization of the setting parameters.

According to the structure described in Figure 1, the yarn is produced by applying both, wrapping and coating processes. Since the two processes can be carried out independently of each other, the manufacturing can vary in sequence.

In the first manufacturing process (WC), the conductive core material is first wound with the polyester substrate material (W). This is followed by the application of thermochromic pigment dispersion (C). In the second manufacturing process (CW), the texturized polyester yarn is first coated and then wound around the stainless steel fiber. The process parameters remain the same for both production methods.

For the wrapping of the stainless steel yarn with the textured polyester, a take-off and rotational speed of 5000 rpm and 1900 tpm is set. The result is a uniform wrapping without gaps or overlaps of the substrate material.

#### 2.2.3. Characterization Methods

Different methods were applied for optical, sensory, colorimetric, and stability characterization and measurement of the heating function.

The optical characterization was carried out with a VHX-600 optical microscope from Keyence (Mechelen, Belgium). The VH-Z20R objective allows a magnification of the sample from 20 to 200. The microscopic images can be used to characterize the homogeneity of spinning and coating. Moreover, confocal laser scanning microscopy images (Keyence VK-X100 series) and SEM images (TM400Plus, Hitachi, Tokyo, Japan) were taken to further characterize the surface morphology and also a cross-section of the yarns.

With a climate chamber HCP from Memmert (Schwabach, Germany), the temperature-sensing function of the yarn was characterized. With the AtmoControl program, the temperature is controlled in a range between 20 and 60 °C. The temperature is increased step by step with an interval of 10 °C. A retention time of 30 min guarantees that the climatic chamber actually reaches the temperature and that the stainless steel yarn can adjust to the temperature. The temperature measurement is carried out on the pure stainless steel yarn, as well as on the coated yarn, and repeated five times each. The yarn is cut to a length of 30 cm, and a multimeter is connected to the ends.

For the characterization of the thermochromic properties of the yarn, measurements to determine the switching temperature, the color change, and the adopted color gradations between switching and room temperature are relevant. The measurements were performed using a Datacolor High-Performance Spectrophotometer 400TM. To analyze the colors, a measuring geometry of d/8°, a 10° observer, and a small measuring aperture (SAV) were applied, and no gloss was used. For the color change, a defined voltage was applied to the yarn with a laboratory power supply unit of the company OJE (number: OJ3005E III) and heated. The color-changing heating yarn is woven in a fabric with a dimension of 5*3 cm. This is positioned under the high-power spectrophotometer with the temperature probe of the multimeter and the connections of the laboratory power supply unit. The laboratory power supply heats the fabric by increasing the supplied voltage step by step, and the resulting temperature is displayed by the multimeter with an integrated FLIR thermal imager. According to the manufacturer, the color gradient should take place at 31 °C. By gradually increasing the temperature by 1 °C, the color gradient between the room temperature and the switching temperature can be characterized. The measurement is repeated three times at different points on a hand-woven fabric made solely from the hybrid yarn.

The heating function of the yarn was characterized by a FLIR thermal imaging multimeter, DM285. With an integrated thermal imaging camera having a resolution of 120 × 160 pixels and an infrared-controlled measuring aid technology, it enables temperature measurement and recording. To heat the yarn, it was connected to the OJE laboratory power supply unit. The hybrid yarn was cut to a length of 30 cm for the measurement. The power supply unit was connected to the ends of the conductive core material and was heated in a controlled manner. For this experiment, different voltages were used, and the temperature of the yarn was measured. The longer the heating yarn, the higher the applied voltage must be. Experiments were carried out with voltages of 1.01 V (0.189 A), 1.67 V (0.322 A), and 3.01 V (0.586 A). Also, the voltage was held constant at 2.10 V, and the time was measured that it takes the yarn to heat up to 40°.

Moreover, textile tests, such as tensile and washing tests, were carried out to determine the mechanical stability of the yarn. The former was performed on the tensile testing machine of the company Zwick Roell (Universal testing machine Zwick 1455, Ulm, Germany) according to EN ISO 2062. Hereby, the hybrid wrapped and coated yarn was compared to the untreated Bekinox fiber. Washing tests were carried out 3 times, and the samples were compared to their untreated equivalents in terms of optic, electrical resistance, heating performance, and color change. For washing, yarn pieces with a length of 50 cm were sewn onto a piece of PES cloth. This cloth was packed into the washing machine IPSO HC60 of the company Treysse and filled with more of the PES cloth. A household washing cycle at 40 °C was performed, and after each washing cycle, the samples were air-dried.

## 3. Results and Discussion

### 3.1. Production of the Temperature-Sensing Heating Yarn with Color Change

During the coating process, the parameters viscosity, fiber material, and take-up speed have an influence on the homogeneity of the coating. In coating trials, different PU-60 dispersions were used, which differ in the amount of thickener. It was found that dispersions with more than 1.35% thickener stuck to the walls of the coating gland and thus led to an inhomogeneous coating. Below 1.35% thickener, the coating flows downwards, despite the upward movement of the yarn. Again, too low viscosity with 1.08% thickener is not recommended as it will run out of the coating nozzle. Further experiments were thus carried out with 1.35% thickener. Additional trials were conducted with different fiber materials, such as cotton, rotor spun, and texturized polyester.

It was found that cotton yarn, in contrast to polyester yarn, absorbs less coating, characterized by low color intensity. At the same time, the texturizing of the polyester yarn leads to a higher uptake of the coating, as it can also be deposited between the individual filaments. Although a higher take-up speed leads to higher take-up of the coating, the polyester yarn requires a rather low take-up speed of about 1.5 m/min to allow drying at a temperature of 140 °C. In Figure 4, the textured polyester can be seen with and without coating.

Figure 5 shows the yarns of the two manufacturing processes, WC and CW. The yarn produced by the first manufacturing route (WC) is less color-intensive since the coating process was conducted in the already wrapped state of the yarn. Hence, less coating can be deposited between the individual filaments.

When the coating is performed after wrapping, the texturing is lost, and only the surface of the polyester yarn is coated. Thus, the second manufacturing process (CW) would be advantageous due to a better coating/color intensity. Nevertheless, the application of the PU60 dispersion changes the feel of the wrapping material. The increased tendency to stick leads to poorer run-off behavior, resulting in many faults with overlapping and missing wrapping. These tests show that the yarn cannot be produced by the second manufacturing route (CW). Thus, the yarn is produced using the first production route (WC) and further characterized.

### 3.2. Characterization of the Temperature-Sensing Heating Yarn with Color Change

There are six methods to characterize the developed temperature-sensing heating yarn with color change. Those methods facilitate the testing and characterization of the sensory, actuator, and heating properties of the yarn.

#### 3.2.1. Optical Characterization

With the microscopic images of the Keyence optical microscope, the construction and homogeneity of the wrapping and coating process can be assessed. Confocal laser scanning microscopy and scanning electron microscopy (SEM) complement this characterization. The images in Figure 6 show that the yarn has a homogeneous coating and wrapping. However, the 50× magnification visualizes that the stainless-steel yarn, despite uniform wrapping, shows through in some places (C1). Figure 7 shows the wrapped and coated yarn in the normal (top) and heated (bottom) conditions. A clearly visible color change from magenta to white takes place.

Figure 8 shows the micrographs of the SEM examination, whereby not only the longitudinal view (left) but also the cross-sectional view (right) was looked at. The cross-sectional micrograph clearly shows the three-layered structure of the hybrid yarn. The steel fiber is recognizable in the core of the yarn, wrapped with fibrous polyester yarn and surrounded by the pigmented coating. The longitudinal view shows no agglomeration of pigments but a rather uniform dispersed distribution.

The morphology of the coating can also be assessed in confocal laser scanning microscopy (Figure 9) and especially in the thereby created 3D image. The 3D image shows the hills and valleys in the coating surface, whereby the amount of height difference of each point compared to the arithmetic mean of the surface (Sa: mean arithmetic height) lies at 29.99 µm.

#### 3.2.2. Sensory Characterization

The sensory function can be characterized by running a climate program in the climate chamber. The resistance of the yarn should change reversibly and linearly corresponding to the temperature change in the environment.

The graph in Figure 10 shows that the stainless steel yarn and the wound and coated yarn run parallel to each other and form a maximum value at 60 °C. The wound and coated yarn have a slightly higher resistance of 0.12 Ω. Consequently, the wrapping has no negative influence on the sensitivity of the core material. Overall, the hybrid yarn has a linear and reproducible resistance curve. The resistance change of 0.23 Ω between 20 and 60 °C is advantageous for the heating function, as this means that the conductivity is not worsened by a higher temperature, and the yarn can be heated evenly and with constant voltage.

#### 3.2.3. Colorimetric Characterization

The color measurement can be used to characterize the thermochromic color change. In this case, the colorimetric measurement is used to evaluate the color intensity of the thermochromic leuco pigments and to analyze the color change from pink to white.

According to the described measurement principle, the color change of the fabric can be described by the reflection values R. Figure 11 shows the reflectance spectrum of the thermochromic pigments between 23 and 35 °C.

As can be seen, there is a stronger reflection in the red than in the green and blue areas (360–600 nm). Large gaps between the curves can be observed, especially at temperatures of 28 and 29 °C. From a temperature of 30 °C onwards, the function takes on a straighter course and no longer changes when the temperature is increased. The color change is complete.

In addition to the reflection spectrum, the color difference DeltaE and the derivation of the amount of substance can be determined. The higher the DeltaE, the bigger the color difference. The DeltaE value is calculated from the temperature-dependent L*a*b values and is calculated as follows [35,36]:(1)$$∆E=\sqrt{{∆L}^{2}+{∆a}^{2}+{∆b}^{2}}$$

Humans can only distinguish colors with a DeltaE value greater than 1. The DeltaE values determined are illustrated in Figure 12. It shows that a color change already takes place between temperatures of 23 and 25 °C and 30 and 35 °C, but people are unable to recognize it. Larger and more visible color differences take place between 27 and 30 °C. Especially between 28 and 29 °C, a clear color change is visible.

#### 3.2.4. Characterization of the Heating Function

According to the described experimental setup, the 30 cm long yarn was heated with voltages of 1.01, 1.67, and 3.01 V. In Figure 13, the thermographic images can be seen. It is visible that the yarn can be heated step by step depending on the length of the yarn and the voltage.

In addition to the heating power at different voltages, it has to be determined how long the yarn needs to cool down again to the ambient temperature of 23 °C after reaching the maximum temperature. For this purpose, a voltage of 2.09 V (0.433 A) is applied to the 30 cm long yarn and is heated to 45 °C. The measurement is repeated three times, and the average time needed to heat up and cool down the yarn to 45 °C and 23 °C (room temperature), respectively, is calculated.

After conducting the experiment, it can be determined that the yarn has a reproducible heating performance, as it heats up to a temperature of 45 °C within 28 s. After switching off the voltage, the heating yarn returned to the ambient temperature of 23 °C in 23 s.

#### 3.2.5. Tensile Tests of Hybrid Yarn Compared to Untreated Steel Fiber

Figure 14 shows the results of the tensile testing of the stainless steel fiber (Bekinox) compared to the hybrid wrapped and coated yarn. Thereby, the hybrid yarns exhibit slightly deteriorated tensile properties. Whereas the Bekinox fibers break at about 80 N, the hybrid yarns break at about 70 N. Also, elongation is slightly reduced, although the stretch properties are negligible as both yarns do not elongate much. Deteriorated tensile properties can be caused by the mechanical stress during processing; the yarn is not only guided through rollers, but also heated to 140 °C during drying.

#### 3.2.6. Washing Test

Lastly, three consecutive washing cycles are performed. After each cycle, the bleeding of the hybrid yarns is checked visually on the white PES fabric to which the yarns have been sewn. No bleeding could be observed; damage to the coating was also not visible. Light microscope images compare the unwashed and washed samples (Figure 15).

Moreover, the electrical resistance before and after washing is measured on 30 cm long samples. No significant differences can be observed, and for the unwashed samples, electrical resistance lies at 4.24 Ω, and for the washed samples, at 4.22 Ω (Table 4).

Also, the heating and color change function is controlled after washing. Therefore, the yarns are heated to 40 °C with 2.10 V, and the time needed to conduct this is measured. For the unwashed samples, it takes 8.28 s to heat up to 40 °C; this time increases to 10.92 s when the samples are washed. The color change is still clearly visible, and the yarns switch from magenta to white. Overall, household washing (three cycles) does not damage the yarns, neither the steel core fiber nor the pigmented coating.

## 4. Conclusions and Outlook

In this paper, a hybrid yarn with a color-changing mechanism is presented. The herein-developed temperature-sensing hybrid heating yarn with a color-changing function offers the possibility to measure the body temperature or actively warm the body by integrating it into textiles. Moreover, the thermochromic color change indicates when a certain temperature has been reached and can thus visually alert the wearer to changes in temperature. The easy manufacturing process that employs purely textile technologies allows the simple upscaling of the process and could be adapted by conventional textile companies. Moreover, the yarn-based design allows the simple integration of the hybrid yarn by common textile technologies (weaving, knitting, sewing). Optimization could be carried out regarding the coating process by employing, e.g., a slot die coating nozzle that applies the coating material with constant pressure.

Due to the thermoresistive effect, the conductive metallic core fiber can represent temperature changes through the measured variable of resistance. At the same time, it may serve as a heating element by connecting it to a voltage source. Both functions could be proven successful and can easily be tuned depending on the application. The core material is wrapped with a textured polyester yarn using the hollow spindle spinning technology. This wrapped polyester yarn serves as the substrate material for the thermochromic coating. By means of vertical dip coating technology, it is possible to apply the thermochromic pigments dispersed in a polyurethane coating. Production specifications and material compositions have been extensively studied, and the best production method could be established.

The hybrid yarn has an excellent heating capacity. A 30 cm long piece heats up to 45 °C within 28 s and returns to room temperature of 23 °C within 23 s. At the same time, the yarn can map temperature changes by measuring resistance and thus also functions as a sensor. The spectroscopic analysis has shown that a clearly visible color change takes place between 28 and 29 °C, which may serve as a visual feedback mechanism. Household washing does not have an impact on the outer appearance, nor on the performance of the yarns.

The hybrid character of the yarn complicates sustainability issues. However, the winding allows at least the unwinding and, thereby, separation of metal and polymeric material. In this way, expensive and precious metal fiber can be reused. Separation of the coating and the PET fiber represents a difficulty.

Regarding future applications, factors such as skin compatibility or abrasion resistance need to be considered and tested if needed. In this way, an additional, transparent, and durable coating is conceivable, which serves as insulation but also protection. This additional layer can protect the underlying functional material from external influences depending on the application (e.g., alkali resistance, abrasion resistance). Above that, different colored yarns can be developed, varying the color of the substrate material and coating, which opens up many design possibilities.

## Figures and Tables

**Figure 1 sensors-23-07076-f001:**
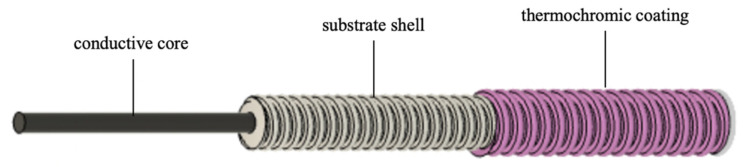
Schematic structure of the temperature-sensing heating yarn with color change.

**Figure 2 sensors-23-07076-f002:**
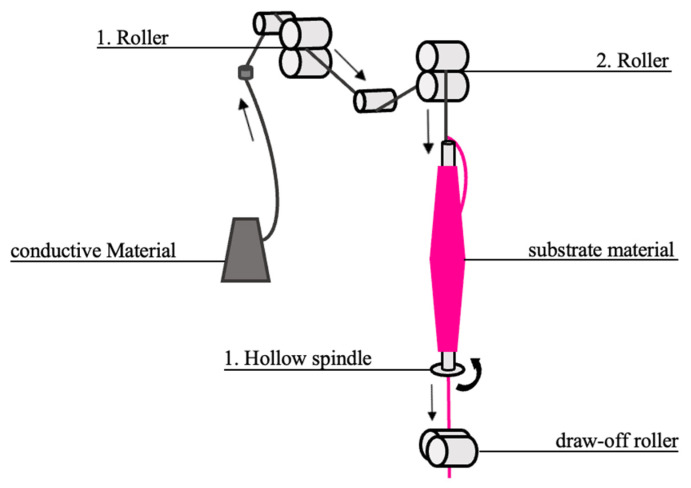
Schematic representation of the thread guide on the rewinding machine.

**Figure 3 sensors-23-07076-f003:**
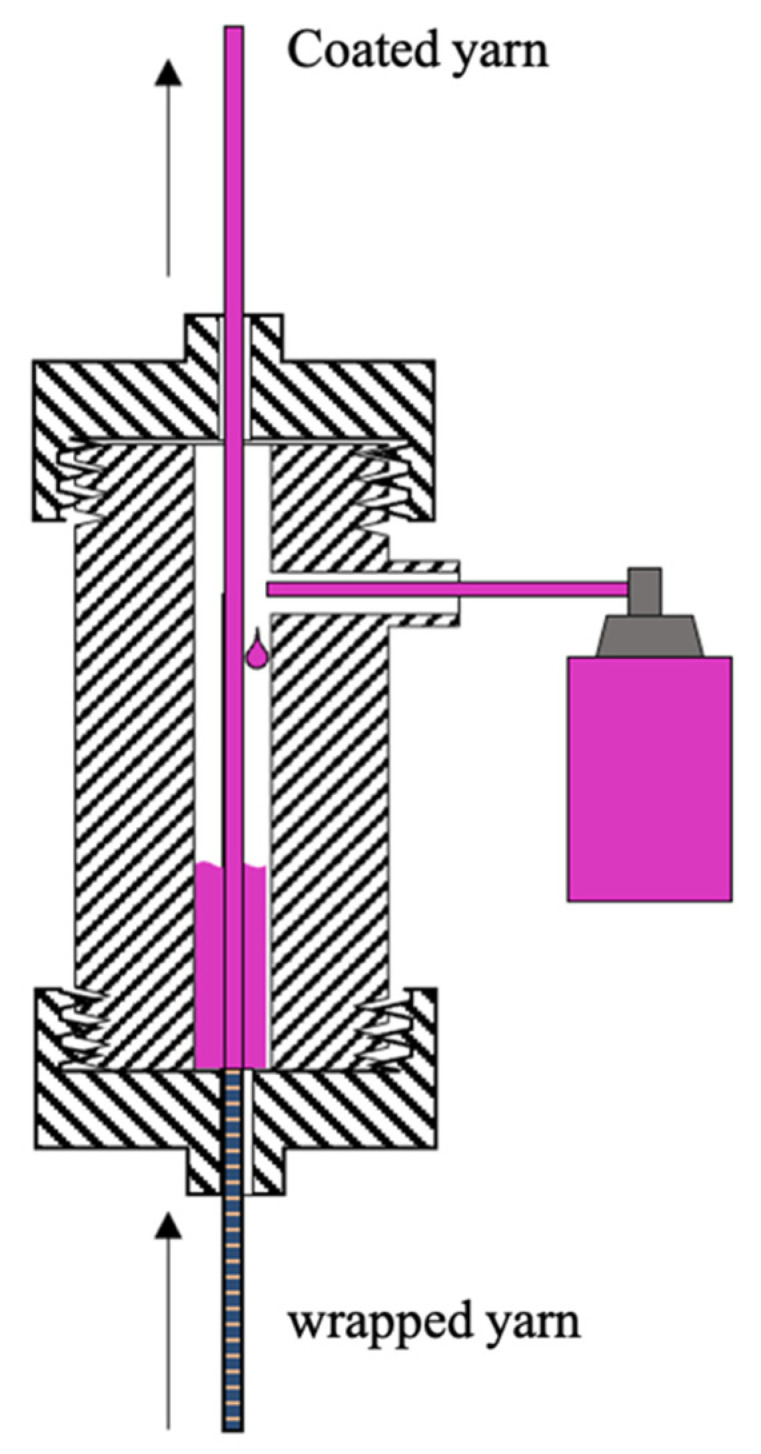
Schematic illustration of the coating process.

**Figure 4 sensors-23-07076-f004:**
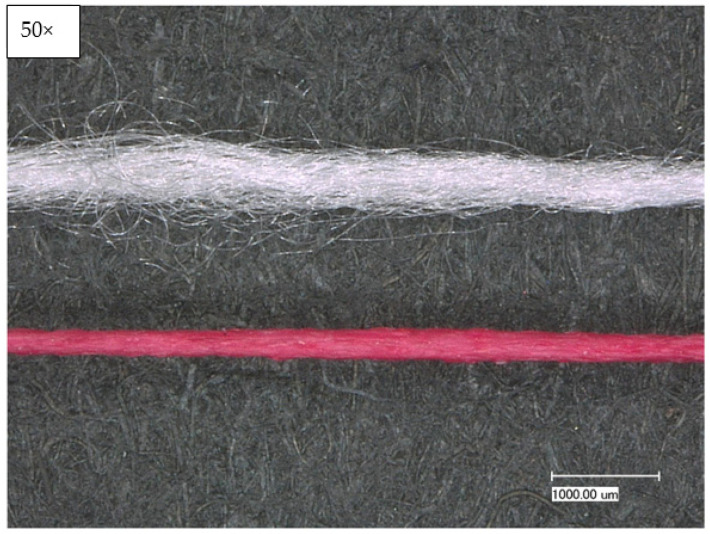
Illustration of the coated polyester yarn; **top**: uncoated textured polyester; **bottom**: coated textured polyester.

**Figure 5 sensors-23-07076-f005:**
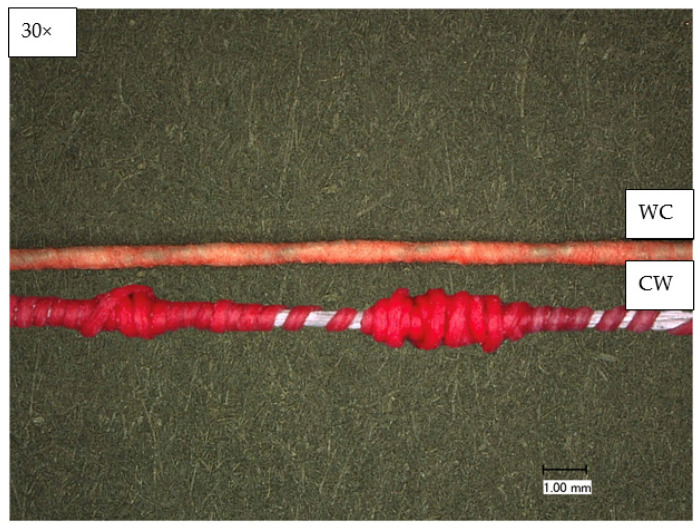
Microscopic picture of the temperature-sensing heating yarn with color change; **top**: produced with the first production route (WC); **bottom**: produced with the second production route (CW).

**Figure 6 sensors-23-07076-f006:**
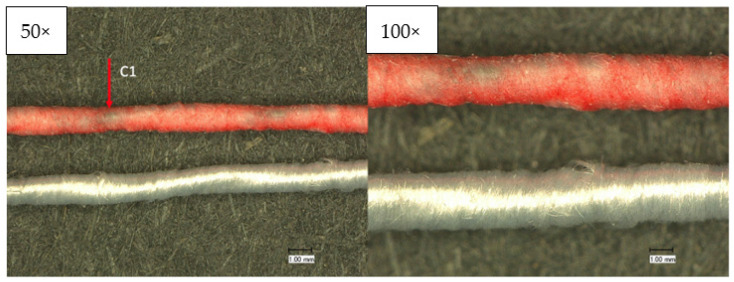
Optical characterization of the temperature sensing heating yarn; **top**: wrapped and coated yarn; **bottom**: wrapped yarn.

**Figure 7 sensors-23-07076-f007:**
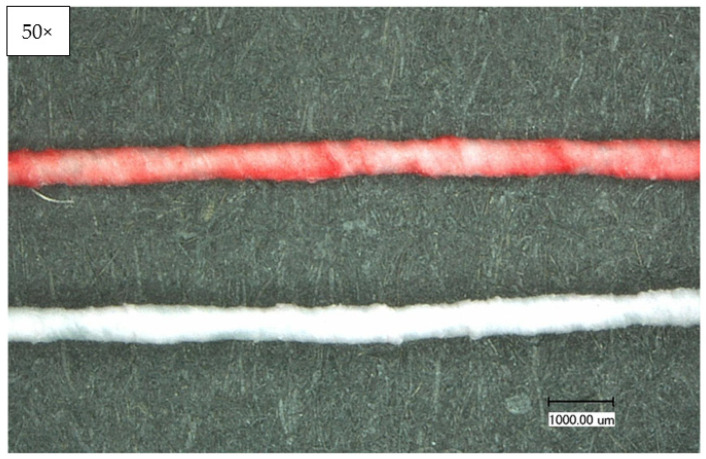
Optical characterization of the temperature sensing heating yarn with color change; **top**: normal condition, **bottom**: heated condition.

**Figure 8 sensors-23-07076-f008:**
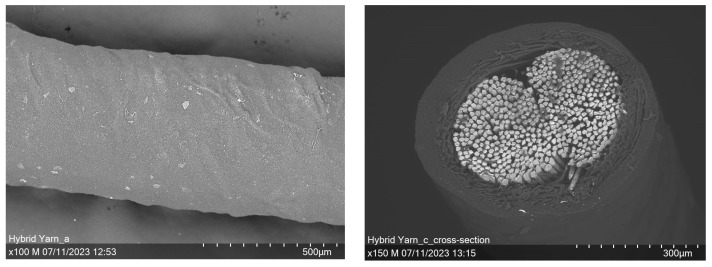
SEM examination of the temperature sensing heating yarn; (**left**) longitudinal view, (**right**) cross-sectional view.

**Figure 9 sensors-23-07076-f009:**
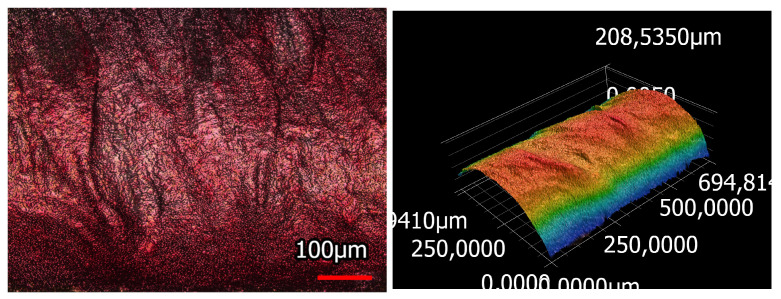
Laser microscopic examination of the temperature sensing heating yarn; (**left**) laser and light microscopic image; (**right**) 3D image of the surface morphology of the coated yarn.

**Figure 10 sensors-23-07076-f010:**
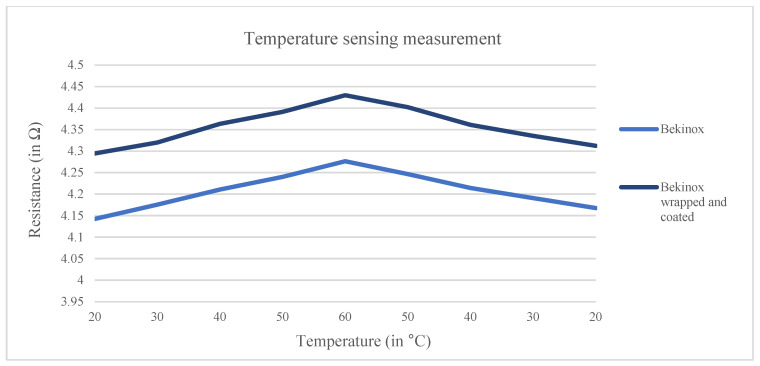
Temperature sensing measurement.

**Figure 11 sensors-23-07076-f011:**
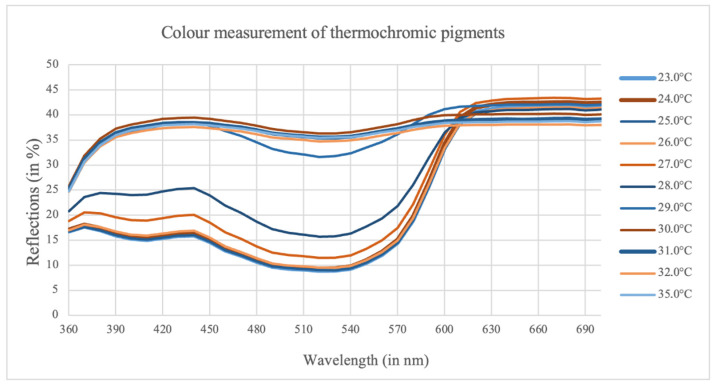
Color measurements of thermochromic pigments.

**Figure 12 sensors-23-07076-f012:**
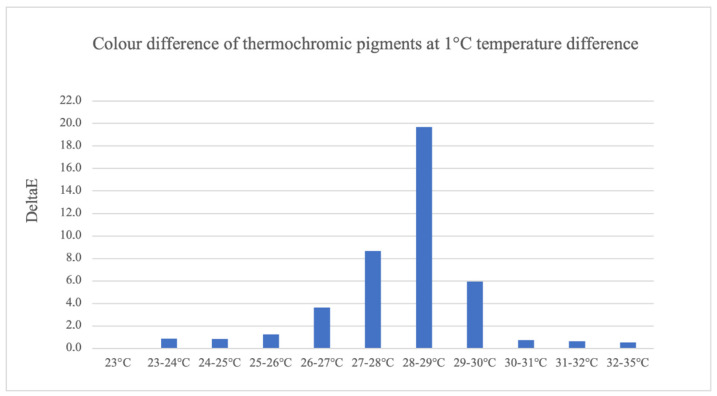
The color difference of thermochromic pigments is dependent on temperature.

**Figure 13 sensors-23-07076-f013:**
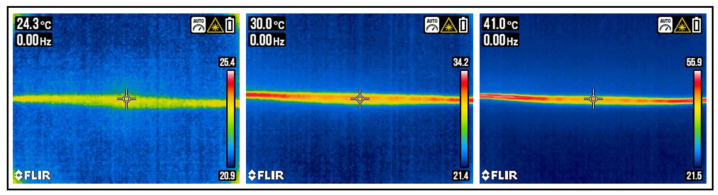
Heated yarn with different voltage levels; (**left**) yarn heated with 1.01 V, (**middle**) yarn heated with 1.67 V, (**right**) yarn heated with 3.01 V.

**Figure 14 sensors-23-07076-f014:**
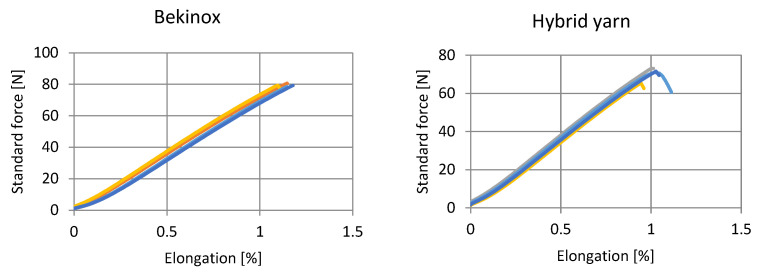
Tensile testing with untreated steel fiber (**left**) and hybrid wrapped and coated yarn (**right**).

**Figure 15 sensors-23-07076-f015:**
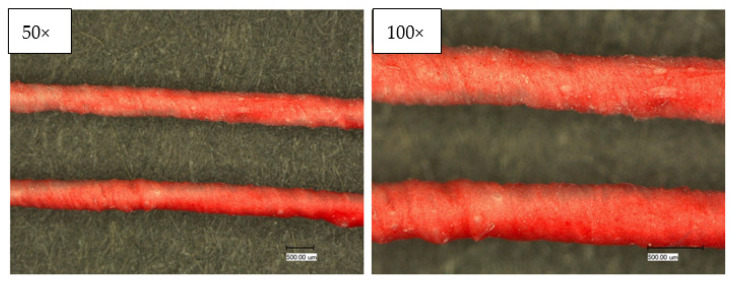
Light microscopic images before (**top**) and after (**bottom**) 3 consecutive washing cycles; (**Left**) 50× magnification, (**right**) 100× magnification.

**Table 1 sensors-23-07076-t001:** Used materials and their specific properties.

Labelling/Trade Names	Material	Manufacturer	Fineness	Specific Properties
Bekinox VN 12/2x275/175S	Stainless steel	Bekaert	505Tex	14.41 Ω/m
Rotor yarn	100% cotton	Schlafhorst	unknown	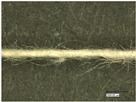
Rotor yarn	100% polyester	Wagenfelder Spinnereien GmbH	10dTex	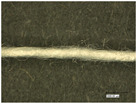
Texturized yarn	100% polyester	Saurer	100f33	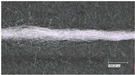
Tubicoat PU60	Polyurethane coating	CHT Germany GmbH	/	White, low viscosity dispersion
Tubicoat Thickener LP	Polyacrylic acid	CHT Germany GmbH	/	/
Thermochromic pigments		Kremer Pigmente GmbH & Co.KG	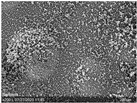	Magenta changes to white at a temperature of 31 °C.

**Table 2 sensors-23-07076-t002:** Coating formulation of the color-changing layer.

Materials	Formula 1 (in %)	Formula 2 (in %)	Formula 3 (in %)	Formula 4 (in %)
Deionized water	64.86	64.7	64.52	64.34
Tubicoat thickener LP	1.08	1.35	1.61	1.88
Tubicoat PU60	32.43	32.35	32.26	32.17
Thermochromic pigments	1.62	1.62	1.61	1.61

**Table 3 sensors-23-07076-t003:** Parameters for characterizing the setting parameters in the coating process.

Influence Parameter	Constant	Variable
Viscosity of the coating	The coating on cotton rotor spun yarn; Drying temperature: 130 °C; Take-up speed 1.25 m/min.	Four Pastes according to the formulas in Table 2 with: 1.08% thickener; 1.35% thickener; 1.61% thickener; 1.88% thickener.
Take-up speed	The coating on cotton rotor spun yarn; Coating with 1.35% thickener; Drying temperature: 90 °C.	The range between 0.5 and 4 m/min: Speed 1: 0.5 m/min; Speed 2: 1.0 m/min; Speed 3: 1.5 m/min; Speed 4: 2.0 m/min; Speed 5: 4.0 m/min.
Fiber material	Coating with 1.35% thickener; The drying temperature of cotton yarn: 90 °C; The drying temperature of polyester yarns: 135 °C; Take-off speed: 1.5 m/min.	Three different yarns: Rotor-spun cotton; Rotor-spun polyester; Textured polyester.

**Table 4 sensors-23-07076-t004:** The electrical resistance of hybrid yarn before and after washing.

	Unwashed	Washed
Sample 1	4.30 Ω	4.26 Ω
Sample 2	4.23 Ω	4.23 Ω
Sample 3	4.19 Ω	4.16 Ω

## Data Availability

Not applicable.

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
