# Peer review of "Development and Characterization of Hybrid, Temperature Sensing and Heating Yarns with Color Change"

_sensors, 2023, doi:10.3390/s23167076_

Round 1

Reviewer 1 Report

Manuscript ID: sensors-2423774

Title: Development and Characterization of Hybrid, Temperature Sensing and Heating Yarns with Colour Change

The paper entitled “Development and Characterization of Hybrid, Temperature Sensing and Heating Yarns with Colour Change” by Theresa Junge and his coworker reported an interesting experimental and developing discussion in the a hybrid yarn, which can measure and visualise temperature changes through a thermoresistive and thermochromic effect. The finding is significant in manipulating the fiber technology for textile area. Although the paper is not a pioneer work in thermochromic materials, it demonstrates the use of simple temperature-sensing hybrid heating yarn with colour-changing function offers the possibility to measure the body temperature or actively warm the body by integrating it into textiles.

This manuscript should be revised according to the following comments:

1. In the application of textile fibers, the test of washing fastness is very important. Because it can be evaluated that this material is applicable. Therefore, it is suggested that relevant experiments should be carried out according to the AATCC test specification.

2. Continuation of the previous question. Is there an assessment of the acid and alkali resistance of sweat?

3. For the part of thermochromic pigments, it is suggested that the characteristics of nano-dispersion of pigments should be explored. And further observe whether the pigments on the surface of the fiber have good dispersion morphology and characteristics.

4. The morphology of pigment nanoparticles and PET is important in the dispersion technology. It is suggested that more detailed microscopes can be used to understand the interaction forces between pigment nanoparticles and PET yarns surface, such as SEM or AFM.

5. Textile materials require exploration of mechanical physics. A universal tensile machine test is recommended.

6. Literature citations are too old. Literature citations need to be fully updated.

Overall, it is recommended for publication in the Sensors after the major revisions.

Author Response

Response to Reviewer 1 Comments

The paper entitled “Development and Characterization of Hybrid, Temperature Sensing and Heating Yarns with Colour Change” by Theresa Junge and his coworker reported an interesting experimental and developing discussion in the a hybrid yarn, which can measure and visualise temperature changes through a thermoresistive and thermochromic effect. The finding is significant in manipulating the fiber technology for textile area. Although the paper is not a pioneer work in thermochromic materials, it demonstrates the use of simple temperature-sensing hybrid heating yarn with colour-changingfunction offers the possibility to measure the body temperature or actively warm the body by integrating it into textiles.

Dear Reviewer, 

Thanks you for your honest feedback and insightful input. We tried our best to include most of your aspects in that little amount of time. In this way, we hope you will find the points you missed so far and like our paper better. 

I will now answer your comments point by point and explain the changes we made.

Thank you for taking the time reading our contribution and best regards
Theresa Junge

Point 1: In the application of textile fibers, the test of washing fastness is very important. Because it can be evaluated that this material is applicable. Therefore, it is suggested that relevant experiments should be carried out according to the AATCC test specification.
Response 1: We carried out 3 consecutive washing cycles (household washing at 40°C) and tested the performance and look of the yarn afterwards.

Point 2: Continuation of the previous question. Is there an assessment of the acid and alkali resistance of sweat?
Response 2: Depending on the application, it would definitely make sense to test the alkali/acid resistance. Up to now, we neglected the assessment of acid and alkali resistance but included an outlook in the discussion concerning future improvements. An idea would be to apply an additional coating as protective and insulating layer.

Point 3: For the part of thermochromic pigments, it is suggested that the characteristics of nano-dispersion of pigments should be explored. And further observe whether the pigments on the surface of the fiber have good dispersion morphology and characteristics.
Response 3: For that we included SEM and laser microscope images that map the surface of the coating.

Point 4: The morphology of pigment nanoparticles and PET is important in the dispersion technology. It is suggested that more detailed microscopes can be used to understand the interaction forces between pigment nanoparticles and PET yarns surface, such as SEM or AFM.
Response 4: We included additional laser mircoscope and SEM images. 

Point 5: Textile materials require exploration of mechanical physics. A universal tensile machine test is recommended.
Response 5: We included a tensile test. Wrapping and coating does not change the tensile properties much. 

Point 6: Literature citations are too old. Literature citations need to be fully updated.
Response 6: In the introduction, we included some more up-to-date literature. 

Overall, it is recommended for publication in the Sensors after the major revisions.

Reviewer 2 Report

Journal: Sensors (ISSN 1424-8220)

Manuscript ID: sensors-2423774

Type: Article

Title: Development and Characterization of Hybrid, Temperature Sensing and Heating Yarns with Colour Change

Summary of the Manuscript

The authors reported on the development and characterization of a hybrid, temperature-sensing and heating yarn with color change. They described a production process involving wrapping and coating to create the yarn structure. They highlighted that the optical characterization showed uniform wrapping and coating, while the sensory characterization demonstrated a linear and reproducible resistance curve for temperature sensing.

Comments

1.     Novelty: The novelty of the study should be emphasized more explicitly, given that there are a lot of similar works in the literature, which can be verified with a simple Google search.

 2.     Introduction: The article lacks a clear and concise introduction that provides a comprehensive overview of the motivation behind developing temperature-sensing and heating yarns with color change. Can you provide more background information on the potential applications and benefits of such yarns?

 3.     Materials and Methods: The article briefly mentions the materials used in the manufacturing process, such as conductive core material, polyester substrate, and thermochromic pigment dispersion. However, it lacks detailed information on the specific properties and characteristics of these materials. Could you provide more details about the materials used and their selection criteria? There are also many ideas in the Result and Discussion Section that should be brought here.

4.     Manufacturing Process: The article mentions the combination of wrapping and coating processes to produce the yarn. However, it lacks a step-by-step description of these processes. Could you provide a detailed explanation of the wrapping and coating processes, including the equipment used, process parameters, and any challenges encountered during manufacturing?

5.     Fabrication Process Reproducibility: The article mentions the fabrication process of the yarn but lacks information on the reproducibility of the process. Have multiple batches of yarns been produced using the same process, and if so, how consistent are the resulting properties? What statistical tool was used to determine if the results were significantly different or not?

  6.     Characterization Methods: The article describes four methods for characterizing the developed yarns. However, it does not provide sufficient information on the specific techniques or instruments used for optical characterization, sensory characterization, colorimetric characterization, and characterization of the heating function. Please provide more details on the measurement techniques and instruments used for each characterization method.

7.     There are many ideas in the Result and Discussion Section that should be taken to the Materials and Methods Section. Overall, the result was not supported by statistical tools.

8.     Coating Durability: The durability of the coating is an important aspect to consider, especially when the yarn is subjected to wear and washing. Has the durability of the coating been evaluated? Are there any concerns regarding the stability and longevity of the color change effect?

9.     Heating Efficiency: The article briefly mentions the heating function of the yarn at different voltages. Could you provide more details on the heating efficiency, such as the heating rate and the power consumption of the yarn? How does the heating performance compare to existing heating technologies?

10.  Safety Considerations: As the yarn is designed for heating purposes, safety is of utmost importance. Has the article addressed any safety concerns associated with the yarn, such as overheating, electrical insulation, or potential hazards when in contact with the skin?

11.  Scalability and Industrial Application: The article focuses on the development and characterization of the yarn but lacks information on its scalability and potential industrial applications. Are there any limitations or challenges in scaling up the production process? Have potential applications, such as in smart textiles or medical devices, been discussed?

12.  Environmental Impact: Considering the increasing importance of sustainability, it would be valuable to address the environmental impact of the developed yarn. Were any eco-friendly materials or manufacturing processes considered during development? Are there any plans for further optimization to reduce the environmental footprint?

13.  Longevity of Sensing Function: The article briefly mentions the sensory function of the yarn based on resistance measurements. However, it does not provide information on the long-term stability and reliability of the sensing function. Has the yarn's sensing capability been tested over an extended period of use? Is there any degradation or drift in the sensing properties?

 14.  Sensing Range: It would be helpful to know the temperature range over which the yarn exhibits accurate and reliable sensing. Are there any limitations or challenges in terms of the upper and lower temperature limits for accurate temperature sensing?

15.  Integration Challenges: The article mentions the hybrid nature of the yarn, combining different materials and functionalities. Were there any challenges or limitations in integrating these components into a single yarn? How does the integration process impact the mechanical properties and flexibility of the yarn?

16.  Comparison with Existing Technologies: To provide a comprehensive understanding, it would be beneficial to compare the developed yarn with existing temperature-sensing and heating technologies. How does the yarn's performance, ease of use, and cost compare to other available options on the market?

17.  Conclusion and Future Directions: The article lacks a conclusive summary and future directions for further research and development. It would be beneficial to include a section that highlights the main findings, potential improvements, and future directions in terms of optimizing the manufacturing process, enhancing performance, and exploring new applications.

Addressing these concerns and providing additional information would greatly enhance the clarity, comprehensiveness, and impact of the article.

Author Response

Response to Reviewer 2 Comments

The authors reported on the development and characterization of a hybrid, temperature-sensing and heating yarn with color change. They described a production process involving wrapping and coating to create the yarn structure. They highlighted that the optical characterization showed uniform wrapping and coating, while the sensory characterization demonstrated a linear and reproducible resistance curve for temperature sensing. 

Dear Reviewer, 

Thanks you for your honest feedback and insightful input. We tried our best to include most of your aspects in that little amount of time. In this way, we hope you will find the points you missed so far and like our paper better. 
I will now answer your comments point by point and explain the changes we made.

Thank you for taking the time reading our contribution and best regards
Theresa Junge

Point 1: Novelty: The novelty of the study should be emphasized more explicitly, given that there are a lot of similar works in the literature, which can be verified with a simple Google search. 
Response 1: We revised the introduction. In this way, we hope to clarify the novelty of the work. 

Point 2: Introduction: The article lacks a clear and concise introduction that provides a comprehensive overview of the motivation behind developing temperature-sensing and heating yarns with color change. Can you provide more background information on the potential applications and benefits of such yarns?
Response 2: We added some more literature in the introduction and tried to emphasize, why flexible textile heating and temperature sensing elements are of interest.

Point 3: Materials and Methods: The article briefly mentions the materials used in the manufacturing process, such as conductive core material, polyester substrate, and thermochromic pigment dispersion. However, it lacks detailed information on the specific properties and characteristics of these materials. Could you provide more details about the materials used and their selection criteria? There are also many ideas in the Result and Discussion Section that should be brought here.
Response 3: We included an additional table that sums up the materials used and their specific properties.

Point 4: Manufacturing Process: The article mentions the combination of wrapping and coating processes to produce the yarn. However, it lacks a step-by-step description of these processes. Could you provide a detailed explanation of the wrapping and coating processes, including the equipment used, process parameters, and any challenges encountered during manufacturing?
Response 4: In our point of view, we have given a detailed description of the manufacturing method. If more information is needed, interested persons should feel free to ask us. But for the understanding of the paper more information on the processing is not necessary.  

Point 5: Fabrication Process Reproducibility: The article mentions the fabrication process of the yarn but lacks information on the reproducibility of the process. Have multiple batches of yarns been produced using the same process, and if so, how consistent are the resulting properties? What statistical tool was used to determine if the results were significantly different or not?
Response 5: Multiple batches have been produced and the processes are reproducible. However, we did not use any statistical tools other than calculation of the mean value, as this is more of a proof of concept. 

Point 6: Characterization Methods: The article describes four methods for characterizing the developed yarns. However, it does not provide sufficient information on the specific techniques or instruments used for optical characterization, sensory characterization, colorimetric characterization, and characterization of the heating function. Please provide more details on the measurement techniques and instruments used for each characterization method.
Response 6: We checked and all instruments are clearly mentioned and named. As the machines/instruments (e.g. light microscope, climate chamber) used are not complicated to run, no further information is given here, as it is not necessary for understanding the message of the paper.

Point 7: There are many ideas in the Result and Discussion Section that should be taken to the Materials and Methods Section. Overall, the result was not supported by statistical tools.
Response 7: We revised this section and put the relevant parts in the materials and methods section. Statistics, other than mean value, were not applied as this is more of a proof of concept.

Point 8: Coating Durability: The durability of the coating is an important aspect to consider, especially when the yarn is subjected to wear and washing. Has the durability of the coating been evaluated? Are there any concerns regarding the stability and longevity of the color change effect?
Response 8: Washing tests were carried out to determine durability. Also in the conclusion we discussed the possibility of an extra coating for protection/insulation.

Point 9: Heating Efficiency: The article briefly mentions the heating function of the yarn at different voltages. Could you provide more details on the heating efficiency, such as the heating rate and the power consumption of the yarn? How does the heating performance compare to existing heating technologies?
Response 9: We did not calculate the heating rate as we did not find a matching formula. It would require additional test, that we were not able to perform in that short amount of time. 

Point 10: Safety Considerations: As the yarn is designed for heating purposes, safety is of utmost importance. Has the article addressed any safety concerns associated with the yarn, such as overheating, electrical insulation, or potential hazards when in contact with the skin?
Response 10: Safety issues are a huge topic and should be considered individually depending on the application. In this way, one can design the yarn application and user oriented taking in account the respective risks.

Point 11: Scalability and Industrial Application: The article focuses on the development and characterization of the yarn but lacks information on its scalability and potential industrial applications. Are there any limitations or challenges in scaling up the production process? Have potential applications, such as in smart textiles or medical devices, been discussed?
Response 11: Scalability is given as solely textile technologies were used and the production method is not complicated. Moreover, the yarn-based character allows the integration by common textile technologies (weaving, knitting, sewing, etc). Potential applications are manifold and reach from automotive to personal protective equipment to medicine. Actually everywhere, were thermal comfort is of interest. 

Point 12: Environmental Impact: Considering the increasing importance of sustainability, it would be valuable to address the environmental impact of the developed yarn. Were any eco-friendly materials or manufacturing processes considered during development? Are there any plans for further optimization to reduce the environmental footprint?
Response 12: We included a small paragraph considering the sustainability issue of such a hybrid yarn. At least, the winding allows the easy unwinding and in that way the expensive metallic fibre could be reused. 

Point 13: Longevity of Sensing Function: The article briefly mentions the sensory function of the yarn based on resistance measurements. However, it does not provide information on the long-term stability and reliability of the sensing function. Has the yarn's sensing capability been tested over an extended period of use? Is there any degradation or drift in the sensing properties?
Response 13: Durability test have not explicitly been carried out. However, we used 6 months old yarns when we carried out the tests for these revisions. The yarns still perform quite nicely. Unfortunately, we did not design a durability testing method and therefore we cannot provide qualitative results. Also, this would now not fit our deadline.

Point 14: Sensing Range: It would be helpful to know the temperature range over which the yarn exhibits accurate and reliable sensing. Are there any limitations or challenges in terms of the upper and lower temperature limits for accurate temperature sensing?
Response 14: The material stainless steel is characterised by a high temperature resistance. The Bekinox yarn from the Beakeart company was specially produced for heating applications and is therefore characterised by very good durability. Extensive tests to determine the temperature range were not carried out for this reason.

Point 15: Integration Challenges: The article mentions the hybrid nature of the yarn, combining different materials and functionalities. Were there any challenges or limitations in integrating these components into a single yarn? How does the integration process impact the mechanical properties and flexibility of the yarn?
Response 15: We faced some challenges in the order of production. This is described in 3.1. Also not all wrapping yarns are suitable for subsequent coating. This is descried in 3.1 as well.

Point 16: Comparison with Existing Technologies: To provide a comprehensive understanding, it would be beneficial to compare the developed yarn with existing temperature-sensing and heating technologies. How does the yarn's performance, ease of use, and cost compare to other available options on the market?
Response 16: We tried to clarify the novelty. Most flexible heating or temperature sensing elements can only do one of the functions. We designed a yarn that can do both things at the same time and also give direct visual feedback to the user. Also many flexible heating/temperature sensing elements are fabric based. The yarn structure allows easier further processing and saves up material costs/consumption.

Point 17: Conclusion and Future Directions: The article lacks a conclusive summary and future directions for further research and development. It would be beneficial to include a section that highlights the main findings, potential improvements, and future directions in terms of optimizing the manufacturing process, enhancing performance, and exploring newapplications.
Response 17: We did revise the conclusion and have included some of your points. 

Addressing these concerns and providing additional information would greatly enhance the clarity, comprehensiveness, and impact of the article.

Round 2

Reviewer 1 Report

The revised manuscript satisfied my concerns and it is acceptable for publication.